# Ischemic Stroke and Heart Failure: Facts and Numbers. An Update

**DOI:** 10.3390/jcm10051146

**Published:** 2021-03-09

**Authors:** Anush Barkhudaryan, Wolfram Doehner, Nadja Scherbakov

**Affiliations:** 1Department of Cardiology, Clinic of General and Invasive Cardiology, University Hospital No 1, Yerevan State Medical University, Yerevan 0025, Armenia; dran_bar@yahoo.com; 2Cardiovascular Research Institute Basel, University Hospital Basel, 4056 Basel, Switzerland; 3BIH Center for Regenerative Therapies (BCRT), Charité-Universitätsmedizin Berlin, 13353 Berlin, Germany; wolfram.doehner@charite.de; 4Department of Cardiology, Campus Virchow, Charité-Universitätsmedizin Berlin, DZHK (German Center for Cardiovascular Research), Partner Site Berlin, 13353 Berlin, Germany; 5Center for Stroke Research Berlin (CSB), Charité-Universitätsmedizin Berlin, Augustenburger Platz 1, 13353 Berlin, Germany

**Keywords:** HFrEF, HFpEF, ischemic stroke, risk factor, cardio–cerebral interactions, stroke prevention

## Abstract

Heart failure (HF) is a severe clinical syndrome accompanied by a number of comorbidities. Ischemic stroke occurs frequently in patients with HF as a complication of the disease. In the present review, we aimed to summarize the current state of research on the role of cardio–cerebral interactions in the prevalence, etiology, and prognosis of both diseases. The main pathophysiological mechanisms underlying the development of stroke in HF and vice versa are discussed. In addition, we reviewed the results of recent clinical trials investigating the prevalence and prevention of stroke in patients with HF.

## 1. Introduction

Heart failure (HF) is a complex clinical syndrome with a high prevalence of diagnosed cases worldwide, described by ventricular systolic or diastolic dysfunction associated with a high rate of mortality and morbidity. A number of comorbidities, including diabetes mellitus, coronary artery disease (CAD), and cerebrovascular diseases, may complicate the course of HF. Among them, acute ischemic stroke may lead to life-threatening complications such as acute decompensation of chronic HF and have a negative impact on further management, as well as on the clinical outcome of patients suffering from HF. On the other hand, chronic HF is considered one of the major risk factors associated with the occurrence and unfavorable clinical outcome after ischemic stroke [1,2]. 

Recently, a new classification of the European Society of Cardiology (ESC) of the diagnosis of HF based on the left ventricular ejection fraction (LVEF), clinical signs of HF, and structural and functional myocardial changes has received growing attention [3,4]. This classification includes HF with preserved ejection fraction (HEpEF, LVEF > 50%), HF with mid-range ejection fraction (HFmrEF, LVEF 40–49%), and HF with reduced ejection fraction (HFrEF, LEVF < 40%). The current review presents an update on recent epidemiological data, the pathophysiological mechanisms of heart–brain interaction, as well as the advancements in the prevention and treatment of acute ischemic stroke in the setting of chronic HF.

## 2. The Prevalence of Ischemic Stroke in Heart Failure 

With the global prevalence of over 104 million cases in 2017 and with an annual incidence rate of 12 million cases, ischemic stroke is considered one of the global disease burdens and a leading cause of death and disability in adult age [5]. Patients with HF have a 2- to 5-fold increased risk of stroke [6,7,8]. In particular, a higher prevalence of ischemic stroke is observed in patients with chronic HF compared to the general population (8–11% vs. 1%) [9]. The stroke rate increases stepwise from 1.3% to 3.5% dependent on the New York Heart Association (NYHA) functional class of HF. Recent analyses of the Swedish Heart Failure (SwedeHF) Registry study suggested no significant differences in stroke prevalence between the HFrEF, HFmrEF, and HFpEF phenotypes; however, the risk of stroke was higher in older patients and those with atrial fibrillation (AF) [10,11]. This is expectable, since aging is associated with a higher prevalence of cardiac arrhythmias, including AF [12,13]. Depending on study designs and patient cohorts, the presence of AF accounts for 2–29% of stroke without a significant difference between HFpEF and HFrEF [14,15]. Therefore, according to the current ESC Guidelines, the presence of HF and older age (>65 years), evaluated by the CHA2-DS2-VASc score (congestive heart failure; hypertension; age ≥ 75, diabetes mellitus; stroke/transitory ischemic attack (TIA); vascular disease, age 65–74; sex category) [16] with ≥2 points, respectively, requires a prescription of continuous oral anticoagulation in patients with AF [17].

## 3. Etiology of Stroke and Risk Factors in Heart Failure

Previously, several classifications of stroke etiology have been proposed [18]. The common clinical one, the Trial of Org 10172 in Acute Stroke Treatment (TOAST), includes five subtypes of ischemic stroke according to the etiology of stroke: cardioembolic, large-artery atherosclerosis, lacunar or small-vessel occlusion, stroke of other determined etiology, and stroke of undetermined etiology [19]. Among these, the cardioembolic subtype of stroke refers to the classical cardiac origin of embolus due to AF, myocardial infarction, endocarditis, cardiac tumors, or valvular heart disease, and accounts for 20–30% of all ischemic stroke cases [20,21]. In recent years, the subtype of stroke of undetermined etiology has been intensively studied, and an alternative diagnosis of embolic stroke of undetermined source (ESUS) has been proposed [22]. The studies investigating ESUS identified the presence of paroxysmal or persistent AF, patent foramen ovale (PFO), atrial cardiomyopathy, and a large artery plaque underling the development of this type of cryptogenic stoke [23,24,25] (Figure 1).

Previously, a relationship between the subtypes of stroke and HF etiology has been shown [26]. In particular, in patients with sinus rhythm, valvular heart disease and dilated cardiomyopathy were mainly associated with cardioembolic stroke, whereas HF developing due to CAD or arterial hypertension has been related to the lacunar or large artery atherosclerotic strokes. However, the cerebrovascular thromboembolism remains the most frequently observed etiological factor, leading to the development of ischemic stroke in patients with chronic HF, including those with CAD [27,28]. Patients with HFrEF are known to possess a prothrombotic state due to platelet hyperactivity, increased thrombin generation, and impaired fibrinolysis [29,30]. The presence of a hypercoagulable state has also been reported in patients with HFpEF [31]. The endothelial dysfunction observed in patients with chronic HF results in decreased endothelium-derived nitric oxide generation and reduced microvascular reactivity of the myocardium, leading to subendocardial damage that finally promotes the development of thromboembolic complications [5,32,33]. Furthermore, pathological ventricular remodeling, including dilatation of LV and/or left atrium (LA), LV akinesia, dyskinesia, or aneurysm may also contribute to the stasis of blood flow and thus, promote thrombus formation [34,35]. These pathological conditions represent the *Virchow’s triad* of hypercoagulable activation [36,37]. The main pathophysiological mechanisms driving the progression of HFrEF, including activation of sympathetic and renin–angiotensin–aldosterone systems, as well as a systemic inflammation, further increase the risk of stroke in these patients. 

Another subtype of ischemic stroke caused by cerebral hypoperfusion has been previously proposed as a hemodynamic stroke [38]. Several factors, including large artery stenosis and occlusion due to atherosclerotic disease, hypotension, or anemia, may underlie the development of this subtype of stroke [38,39,40]. The decrease in cerebral blood flow may be further compromised due to the reduced cardiac output in patients with HF in the presence of large artery stenosis [41,42]. The cerebral hypoperfusion leads to a decrease in blood flow to the areas of brain supplied by deep arteries lacking collateral flow, which makes them vulnerable to ischemic damage [21], vascular dementia [43,44], silent or subclinical stroke, and cognitive impairment [14,45,46]. 

## 4. Cardio–Cerebral Interactions after Acute Stroke 

The development of cardiac complications in the acute and subacute phases of stroke, including structural myocardial injury, revealed by electrocardiographic changes, elevation of biomarkers, such as cardiac Troponin T (cTnT) or natriuretic peptides, as well as acute coronary syndrome, is frequently observed after stroke and may deteriorate the prognosis of stroke patients [47,48,49]. The Troponin Elevation in Acute Ischemic Stroke (TRELAS) sub-study showed that 25% of patients with acute stroke and elevated plasma cTn levels required a percutaneous coronary intervention (PCI) due to the presence of culprit lesion detected by coronary angiography [50]. The elevated levels of cTn were associated with adverse outcome 3 months after stroke [51]. Furthermore, the development of acute myocardial infarction (AMI) during a short- and long-term follow-up after acute ischemic stroke has been frequently reported [52,53,54]. In particular, the presence of diabetes mellitus, congestive HF, and arterial hypertension in patients with stroke has been associated with the development of AMI 1–5 years after the acute cerebrovascular event and related to an increased post stroke mortality [55]. In addition, Takotsubo cardiomyopathy, with a prevalence of 0.4–1.2%, has been reported after acute ischemic stroke, however more commonly in patients with hemorrhagic stroke [56,57]. Thus, the occurrence of stroke-associated cardiac complications may be summarized under *stroke–heart syndrome* [58].

The pathophysiological mechanisms of stroke–heart syndrome have not been sufficiently investigated yet. One of the underlying pathways is the development of acute autonomic dysfunction. A recent experimental study reported a reduction in LVEF, myocardial fractional shortening, and heart rate after acute cerebral ischemia induced by middle cerebral artery occlusion (MCAO) in mice, which was accompanied by elevation of cTnT levels [59]. In addition, impaired catecholamine homeostasis was confirmed by elevated levels of norepinephrine. The results of a previous clinical study revealed a mechanistic link between the central autonomic dysregulation evaluated by central sleep apnea and peripheral endothelial dysfunction in acute stroke [60]. Interestingly, after 1 year follow-up, a disappearance of the central sleep apnea was observed in patients with normalized endothelial function and left-sided stroke. Indeed, experimental and clinical studies suggest a lateralization of the autonomic regulation with a primary involvement of right insular regions. The role of the insular cortex in the regulation of cardiovascular function has been previously reported, showing that right hemisphere insular cortex stroke leads to an increase in sympathetic activity [61,62]. Another study, investigating the impact of cardiac autonomic tone on the clinical outcome in subacute stroke, showed an association between the depressed heart rate variability and impaired functional outcome after post stroke neurological rehabilitation [63]. Thus, central autonomic dysfunction may contribute to the development of cardiovascular complications in patients with acute stroke leading to an adverse clinical outcome. 

Cardio–cerebral interactions in the setting of HF have also been investigated in previous studies. The main pathophysiological mechanisms, including increased inflammation, cerebral hypoperfusion, neurohormonal activation, and decreased thiamine levels, contribute to the development of *cardiocerebral syndrome* characterized by structural changes of the brain leading to cognitive impairment in patients with HF [41]. An association between brain structural damage and decreased blood flow has been previously reported in patients with HF [64]. In another study, a decrease in gray matter density in the areas of hippocampus, precuneus, and frontomedian cortex and increased N-terminal pro-B-type natriuretic peptide (NT-proBNP) levels have been shown in HFrEF [65]. Thus, the heart–brain interaction in the setting of acute stroke and HF mutually contributes to the development of myocardial dysfunction and cerebral impairment, respectively, requiring the administration of treatment of these complications to improve quality of life, survival, and clinical outcome in these patient cohorts.

## 5. The Impact of Heart Failure and Atrial Fibrillation on the Clinical Outcome of Ischemic Stroke 

### 5.1. Heart Failure

The presence of chronic HF has a prognostic significance in patients with acute stroke. In particular, increased levels of natriuretic peptides indicating the severity of chronic HF and decreased LVEF have been shown to be risk factors for development of stroke in patients with HF without AF [35,66]. In the physiologic condition, an elevated LV filling pressure per se leads to an increased contractility (Frank–Starling mechanism); however, in the setting of HF, this mechanism is impaired, the stroke volume decreases subsequently resulting in cerebral hypoperfusion [1]. In a population-based study, including over 7500 participants, the risk of stroke in patients was particularly high (more than five-fold increase) during the first months after diagnosis of HF but attenuated over time [67]. The findings of another study showed that acute decompensation of chronic HF was regarded as an independent predictor of adverse functional outcome in patients with ischemic stroke [68]. In addition, reduced LVEF has been associated with adverse outcome after cardioembolic stroke [69]. Furthermore, the relationship between decreased LVEF and poor neurological outcome (modified Rankin Scale, mRS ≥ 3) has been previously reported in patients with ischemic stroke [1]. Previous clinical studies have shown that apart from clinically manifested strokes, silent strokes were more prevalent in patients with HF compared to subjects without the disease [70,71]. 

### 5.2. Atrial Fibrillation 

The most frequently observed arrhythmia is a non-valvular AF which increases the risk of stroke by 4-5 times [72]. The presence of AF in chronic HF is considered one of the major risk factors of acute ischemic stroke [73,74] associated with an increased post stroke mortality and physical disability [75,76,77]. In a recent study in patients with acute ischemic stroke, an increased risk of in-hospital mortality in patients with HF, regardless of the coexistence of AF, has been reported. However, the risk of stroke recurrence was doubled in patients with AF and HF, as opposed to patients with only HF [78]. These findings are consistent with the results of previous trials showing a higher risk of stroke recurrence in patients with these comorbidities [79,80]. In the Aliskeren Trial on Acute Heart Failure Outcomes (ASTRONAUT) sub-study, patients with atrial fibrillation/flutter (AFF) and HFrEF hospitalized with acute HF showed significantly higher rates of fatal stroke in contrast to patients without AFF (1.8% vs. 0.3%, *p* = 0.011, respectively) during a 12-month follow-up [14]. Recently, the Atrial Fibrillation Clopidogrel Trial With Irbesartan for Prevention of Vascular Events (ACTIVE-W) in patients with permanent AF showed no differences between patients with HFpEF and HFrEF with regard to the risk of embolic events and stroke, suggesting a minor role of the extent of LV dysfunction in the increased risk of stroke [81].

Thus, despite recent advancements in the medical, interventional, and device therapy in patients with HF and AF, acute ischemic stroke remains a severe complication of this disease, with a potentially devastating impact on physical performance, quality of life, and prognosis of HF patients.

## 6. The Prevention and Treatment of Stroke in Patients with Heart Failure

### 6.1. Stroke Prevention in Heart Failure with Sinus Rhythm 

The prevention of ischemic stroke includes lifestyle and diet modification, increase in physical activity, as well as pharmacological treatment of comorbidities, such as arterial hypertension, hyperlipidemia, atherosclerosis, and chronic HF. Anticoagulation therapy may require particular contemplation in the absence of *AF* considering the increased risk of thromboembolic events in patients with HF [82]. A number of clinical trials investigated the impact of anticoagulants and antiplatelet drugs in the prevention of stroke in patients with HF. Previous studies, including Warfarin versus Aspirin in Reduced Cardiac Ejection Fraction (WARCEF), Warfarin and Antiplatelet Therapy in Chronic Heart Failure (WATCH), Heart Failure Long-term Antithrombotic Study (HELAS), and Warfarin/Aspirin Study in Heart Failure (WASH) evaluated the effect of warfarin versus aspirin on the risk of stroke in HFrEF patients with maintained sinus rhythm [35,83,84,85]. The results of the small-sample WASH and HELAS trials have shown no efficacy of anticoagulant therapy on the composite endpoint of death, stroke, or myocardial infarction in the study patients [86,87]. The WATCH trial showed a reduced risk of ischemic stroke with warfarin (international normalized ratio, INR, of 2.5 to 3.0) compared to aspirin (162 mg once daily), or clopidogrel (75 mg once daily), although this effect was neutralized by an increased risk of bleeding [85]. In the WARCEF trial, a significant reduction in the incidence of ischemic stroke with warfarin (INR of 2.0 to 3.5) compared to aspirin (325 mg per day) has been shown only after four years of therapy. However, the rate of major bleeding events significantly increased throughout the treatment in the warfarin group (adjusted rate ratio 2.05 [95% CI 1.36–3.12], *p* < 0.001) [86]. 

Thus, despite the overall reduction in the risk of thromboembolic events, the beneficial effect of warfarin therapy in patients with HFrEF with sinus rhythm could not be confirmed due to an increased risk of bleeding in the study patients [77,87]. Therefore, routine administration of warfarin in HF patients with maintained sinus rhythm is currently not recommended [3,88]. Acetylsalicylic acid (aspirin) is currently used for the secondary prevention of stroke in HF patients with sinus rhythm [21]. 

The perspective usage of novel oral anticoagulants (NOACs) has been investigated as a therapeutic approach for stroke prevention in patients with HF [22]. The results of the Rivaroxaban in Patients with Heart Failure, Sinus Rhythm, and Coronary Disease (COMMANDER HF) trial enrolling patients with HFrEF showed no benefit of rivaroxaban (2.5 mg twice daily) on top of standard care in the prevention of primary outcome, a composite of all-cause death, myocardial infarction, and stroke [89]. Notably, however, the incidence of all-cause strokes or transitory ischemic attacks (TIA) was reduced by 34% in the rivaroxaban group in a secondary analysis [90]. Again, as in WATCH and WARCEF trials, the increased risk of major bleeding (1.68 [95% CI 1.18–2.39], *p* = 0.003) was observed in the COMMANDER HF trial [91]. The Cardiovascular Outcomes for People Using Anticoagulation Strategies (COMPASS) trial, investigating patients with systemic atherosclerotic disease, revealed a 49% reduction in the relative risk of stroke with rivaroxaban (2.5 mg twice daily) plus aspirin (100 mg once daily) compared to aspirin alone (100 mg once daily). However, the treatment with combined antiplatelet therapy resulted in a 70% increase in major bleeding events in this patient cohort [92]. The prevalence of HF in this clinical trial was 20% in each study group. 

### 6.2. Stroke Prevention in Heart Failure with Atrial Fibrillation 

The main complication of AF apart from hemodynamic dysregulation is a systemic thromboembolic event/cardioembolic stroke. The efficacy of prevention of acute stroke with oral anticoagulant drugs in HF patients with AF has been previously shown [93]. 

These medications are recommended for patients with a CHA_2_DS_2_-VASc score of 2 or more which is used for assessment of stroke risk in patients with AF [17,94]. Currently, there is no conclusive data to administer antithrombotic therapy, oral anticoagulant drugs, or aspirin in the case of a CHA_2_DS_2_-VASc score of 1, although assessment of thromboembolic risk of each patient is recommended [95]. However, about 15% of patients with AF and CHA_2_DS_2_-VASc score of 1 may benefit from OACs [95]. Nonetheless, the duration of AF in these patients should be considered and the decision to prescribe these medications in patients with a low score should be individually adjusted [95,96].

The closure of left atrial appendage (LAA), as an alternative therapeutic method, may be considered in high-risk patients with AF who have contraindications to anticoagulants [97,98]. The previous clinical trials investigating percutaneous LAA closure showed noninferiority of this procedure to OACs (NOACs or Vitamin K antagonists) in the prevention of cardiovascular and neurological complications [99,100]. In addition, the results of the LAARGE (Left-Atrium-Appendage occlude Register-Germany) registry revealed stroke prevention with this interventional method in patients with reduced, mid-range, or preserved LVEF [101]. Currently, several clinical trials are investigating the comparative efficacy of LAA occlusion and OACs in the prevention of cardioembolic events [72,102].

Recently, the Early Treatment of Atrial Fibrillation for Stroke Prevention Trial (EAST-AFNET 4) evaluated patients with AF from whom 28% had stable chronic HF. The findings of this study demonstrated an advantage of the early rhythm control therapy (treatment with antiarrhythmic drugs or AF ablation) compared to the symptomatic treatment of AF (usual care). The study showed a 21% risk reduction for combined primary outcome consisting of cardiovascular death, stroke, and worsening of HF or CAD during the median follow-up of 5 years [103]. Therefore, several therapeutic options including oral anticoagulation, rhythm control therapy, and LAA closure are available for stroke prevention and need to be individually adjusted in patients with HF and AF (Table 1). 

### 6.3. Stroke Treatment in HF

The treatment of patients with acute ischemic stroke should be performed in a stroke unit and, if applicable, with recombinant tissue plasminogen activator (rt-PA) and/or intra-arterial thrombectomy followed by antiplatelet therapy. The decreased therapeutic effect of thrombolysis, a higher risk of systemic embolization of LV thrombi, as well as hypoperfusion-related brain injury may complicate the interventional therapy of acute stroke in patients with HF [21]. However, the findings of a recent study have shown a favorable effect of thrombolysis with rt-PA on stroke outcome disregarding the presence of HF in this study cohort. In addition, a meta-analysis reported a higher risk of bleeding due to thrombolysis in stroke patients with HF (odds ratio 1.96; [95% confidence interval 1.30–2.94]) compared to stroke patients without HF [106]. The presence of congestive HF has also been associated with an unfavorable clinical outcome after an intra-arterial thrombectomy compared to patients without HF [107]. Further clinical trials directed to the study of the efficacy of mechanical recanalization treatment in the setting of HF in patients with acute ischemic stroke are warranted. 

## 7. Conclusions

HF is a severe clinical syndrome associated with a high mortality and morbidity, particularly among the elderly population worldwide. Stroke is considered one of the major comorbidities of HF requiring timely prevention and treatment. The presence of HF is associated with a poor prognosis in stroke survivors. The development of acute ischemic stroke, mainly induced by a prothrombotic state, the presence of AF, and cerebral hypoperfusion, may lead to acute decompensation of chronic HF and is associated with a physical disability and increased post stroke mortality. The coexistence of acute stroke and chronic HF in many patients leads to both exacerbation of cerebral injury and progression of myocardial dysfunction. Recent clinical trials have shown promising results regarding the administration of NOACs for the prevention of stroke in patients with HF. The understanding of complex pathophysiological mechanisms behind stroke–heart syndrome will contribute to the advancement of knowledge in the diagnosis and management of these pathological conditions. Future studies are required to reveal novel therapeutic strategies to improve the quality of life, functional outcome, and survival of HF patients with acute cerebrovascular diseases. 

## Figures and Tables

**Figure 1 jcm-10-01146-f001:**
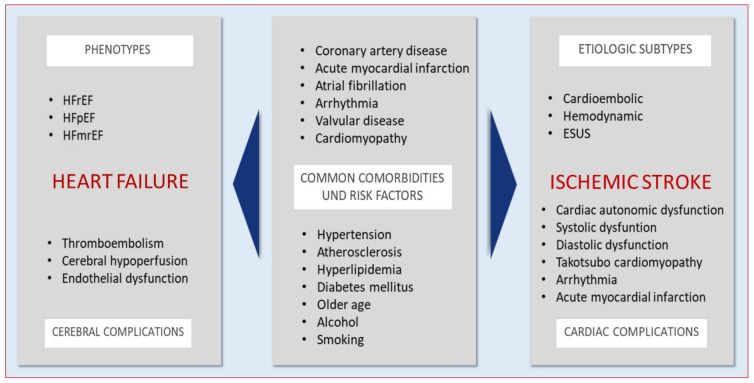
The comorbidities and risk factors in ischemic stroke and heart failure.

**Table 1 jcm-10-01146-t001:** An overview of clinical trials on stroke prevention in heart failure.

Year	Clinical Trial	No. of Patients	Type of HF	Mean LVEF, %	Heart Rhythm	Mean FU, Months	Results
**2004**	**WASH** (Warfarin/Aspirin Study in Heart Failure) [84]	279	HFrEF	≤35	SR	27	No difference in PO (death, nonfatal myocardial infarction, or nonfatal stroke) between warfarin vs. aspirin vs. no antithrombotic treatment
**2006**	**HELAS** (Heart Failure Long-term Antithrombotic Study) [85]	197	HFrEF	<35	SR	22	No difference in PO (nonfatal stroke, peripheral or pulmonary embolism, myocardial re-infarction, re-hospitalization, exacerbation of heart failure, or death from any causes) between aspirin vs. anticoagulant therapy
**2009**	**PROTECT AF** (WATCHMAN left Atrial Appendage System for Embolic Protection in Patients with Atrial Fibrillation) [104]	707	Chronic HF	≥30	AF	48	No difference in PO (stroke, systemic embolism, or cardiovascular/unexplained death) between LAA closure vs. warfarin
**2009**	**WATCH** (Warfarin and Antiplatelet Therapy in Chronic Heart Failure) [83]	1587	HFrEF	<35	SR/AF	21	No difference in PO (all-cause mortality, nonfatal myocardial infarction, nonfatal stroke) between warfarin vs. aspirin vs. clopidogrel. Reduction in strokes but more hemorrhage in warfarin treatment
**2012**	**WARCEF** (Warfarin versus Aspirin in Reduced Cardiac Ejection Fraction) [86]	2305	HFrEF	25	SR	72	No difference in PO (death, ischemic stroke, or intracerebral hemorrhage) between aspirin vs. warfarin. Prevention of ischemic strokes but more major hemorrhage
**2014**	**PREVAIL** (Evaluation of the WATCHMAN LAA Closure Device in Patients With Atrial Fibrillation Versus Long Term Warfarin Therapy) [105]	407	Chronic HF	≥30–<35	AF	48	No difference in PO (stroke, systemic embolism, or cardiovascular/unexplained death) between the LAA closure vs. warfarin
**2015**	**ACTIVE** (Atrial Fibrillation Clopidogrel Trial With Irbesartan for Prevention of Vascular Events) [81]	3487	HFrEF HFpEF	<45	AF	43	No difference in PO (stroke, transient attack and systemic embolism) between HFrEF vs. HFpEF
**2017**	**COMPASS** (Cardiovascular Outcomes for People Using Anticoagulation Strategies) [92]	27,395	HFrEF	NA	AF	23	Improved cardiovascular outcome (cardiovascular death, stroke, myocardial infarction) but more major bleedings in patients with rivaroxaban and aspirin
**2018**	**COMMANDER HF** (Rivaroxaban in Patients with Heart Failure, Sinus Rhythm, and Coronary Disease) [89]	5022	HFrEF	<40	SR	20	No difference in PO (death from any cause, myocardial infarction, or stroke) between treatment with antiplatelet agents and rivaroxaban in addition to antiplatelet agents.
**2020**	**EAST-AFNET 4** (Early Treatment of Atrial Fibrillation for Stroke Prevention Trial) [103]	2789	HFrEF	NA	AFF	61	The prevention of stroke by rhythm-control therapy in patients with AF and cardiovascular conditions
**2020**	**LAARGE** (Left-Atrium-Appendage occlude Register-Germany registry) [101]	619	HFpEF HFmrEF HFrEF	>55 36–55 ≤35	AF	12	The efficacy of LAA closure in the prevention of stroke in patients with AF
**2020**	**PRAGUE-17** (Left Atrial Appendage Closure vs. Novel Anticoagulation Agents in Atrial Fibrillation) [100]	402	Chronic HF	NA	AF	21	No difference in occurrence of stroke and increased risk of bleeding between LAA closure and therapy with direct oral anticoagulant

AFF, atrial fibrillation or flutter; CHF, chronic heart failure; CV, cardiovascular; DOAC, direct oral anticoagulant; HFmrEF, heart failure with mid-range ejection fraction; HFpEF, heart failure with preserved ejection fraction; HFrEF, heart failure with reduced ejection fraction; LAA, left atrial appendage; LVEF, left ventricular ejection fraction; PO, primary outcome; SR, sinus rhythm; TIA, transient ischemic attack.

## Data Availability

Not applicable.

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
