# Peer review of "Ischemic Stroke and Heart Failure: Facts and Numbers. An Update"

_jcm, 2021, doi:10.3390/jcm10051146_

Round 1

Reviewer 1 Report

It is very well written review. Perhaps adding one or two figures may make it more attractive and easy for quick read. 

Consider adding a table mentioning names and key points of most important trials regarding ischemic stroke and heart failure. 

Author Response

Reviewer 1

It is very well written review. Perhaps adding one or two figures may make it more attractive and easy for quick read. 

Consider adding a table mentioning names and key points of most important trials regarding ischemic stroke and heart failure. 

Reply: We would like to thank the reviewer for these comments. Due to limited time, an addition of further figures was not possible. However, we added a Table 1, summarizing the Clinical Trials on stroke prevention in heart failure.

Reviewer 2 Report

Good and comprehensive review by the authors and most topics are well covered and well organized. 

Minor revisions required mostly in terms of simplifying sentence construction to improve readibility

Some additional details required which have been mentioned in the file in order to improve the quality of the guidelines being provided by the authors. 

Reviewer Comments/Recommendations/Queries:

  1. Line 99: The authors suggest large artery occlusion/obstruction without discussing stenosis which is a significant cause for hypo-perfusion in the presence of heart failure. Additionally, obstruction is not a commonly used scientific terminology in neurovascular literature. For complete flow arrest, the reviewer would recommend using occlusion and stenosis for partial flow reduction from atherosclerotic disease.
  2. Line 101: The authors should mention that the low flow state in heart failure leads to compounding of the reduce cerebral blood flow in the presence of narrowing/occlusion
  3. Line 104: The authors should mention vascular dementia in addition to sub clinical stroke as well as cognitive impairment.
  4. Lines 119-121: The authors discussed the stroke-heart syndrome especially in the presence of Takasubo’s cardiomyopathy. However, this is more commonly seen in hemorrhagic stroke which should be mentioned since the mechanism is quite prominent in the setting of subarachnoid hemorrhage
  5. Line 172: The authors mentioned “advance neurological deficit”. However, this is not an established terminology. This reviewer could not understand what this meant: does this imply worsening of outcome, worsening of neurological deficit or an overall poorer functional score for the same level of deficit
  6. Line 177-178: The reviewer recommends using better syntax in this sentence construction.
  7. Line 179: this line is redundant given the details of the authors go into  before and after this specific mention.
  8. Line 224 & 233: The reviewer recommends mentioning the absolute dose/range of doses/target therapeutic range for all three agents (aspirin, warfarin and Rivaroxaban) that are being compared in these studies.
  9. Lines 235-249: The reviewer would like to know if the authors are aware of any data on increase hemorrhagic risk in the trials mentioned and if they do then it should be included in this part of the discussion.
  10. Lines 261-64: The reviewer would recommend simplifying the language in the sentence since it’s quite confusing and difficult to comprehend

Author Response

Reviewer 2

Good and comprehensive review by the authors and most topics are well covered and well organized. 

Minor revisions required mostly in terms of simplifying sentence construction to improve readibility

Some additional details required which have been mentioned in the file in order to improve the quality of the guidelines being provided by the authors. 

Reviewer Comments/Recommendations/Queries:

  1. Line 99: The authors suggest large artery occlusion/obstruction without discussing stenosis which is a significant cause for hypo-perfusion in the presence of heart failure. Additionally, obstruction is not a commonly used scientific terminology in neurovascular literature. For complete flow arrest, the reviewer would recommend using occlusion and stenosis for partial flow reduction from atherosclerotic disease.Replay: We thank the reviewer for this comment. We changed the discussion now:Several factors, including large artery stenosis and occlusion due to atherosclerotic disease, hypotension or anemia, may underlie the development of this subtype of stroke.
  2.  
  3.  
  4.  
  5. Line 101: The authors should mention that the low flow state in heart failure leads to compounding of the reduce cerebral blood flow in the presence of narrowing/occlusionReplay: We thank the reviewer for this comment. We rewrote the discussion:The decrease in cerebral blood flow may be further compromised due to the reduced cardiac output in patients with HF in the presence of large artery stenosis.
  6.  
  7.  
  8.  
  9. Line 104: The authors should mention vascular dementia in addition to sub clinical stroke as well as cognitive impairment.Replay: We thank the reviewer for this comment. We added now:The cerebral hypoperfusion leads to a decrease in blood flow to the areas of brain supplied by deep arteries lacking collateral flow which makes them vulnerable to ischemic damage, vascular dementia, silent or subclinical stroke and cognitive impairment.
  10.  
  11.  
  12.  
  13. Lines 119-121: The authors discussed the stroke-heart syndrome especially in the presence of Takasubo’s cardiomyopathy. However, this is more commonly seen in hemorrhagic stroke which should be mentioned since the mechanism is quite prominent in the setting of subarachnoid hemorrhage   
  14. In addition, Takotsubo cardiomyopathy, with a prevalence of 0.4%-1.2%, has been reported after acute ischemic stroke, however more commonly in patients with hemorrhagic stroke.
  15. Replay: We thank the reviewer for this comment. We added now:
  16.  
  17. Line 172: The authors mentioned “advance neurological deficit”. However, this is not an established terminology. This reviewer could not understand what this meant: does this imply worsening of outcome, worsening of neurological deficit or an overall poorer functional score for the same level of deficitReplay: We thank the reviewer for this comment. Under the ”advanced neurological deficit”, a poor neurological deficit was mentioned. We added now:Furthermore, the relationship between decreased LVEF and poor neurological outcome (modified Rankin Scale, mRS≥3) has been previously reported in patients with ischemic stroke
  18.  
  19.  
  20.  
  21. Line 177-178: The reviewer recommends using better syntax in this sentence construction.Repay: We thank the reviewer for this comment. We rephrased the sentence now:The most frequently observed arrhythmia is a non-valvular AF associated with 4 to 5 increased risk of stroke
  22.  
  23.  
  24.  
  25. Line 179: this line is redundant given the details of the authors go into  before and after this specific mention.Repay: We thank the reviewer for this comment. We rephrased the sentence now:The presence of AF in chronic HF is considered one of the major risk factors of acute ischemic stroke associated with an increased post-stroke mortality and physical disability.
  26.  
  27.  
  28.  
  29. Line 224 & 233: The reviewer recommends mentioning the absolute dose/range of doses/target therapeutic range for all three agents (aspirin, warfarin and Rivaroxaban) that are being compared in these studies.Repay: We thank the reviewer for this important comment. We added the doses for all three agents mentioned in these studies:The WATCH trial showed a reduced risk of ischemic stroke with warfarin (international normalized ratio, INR, of 2.5 to 3.0) compared to aspirin (162 mg once daily), or clopidogrel (75 mg once daily), although this effect was neutralized by an increased risk of bleeding [82]. In the WARCEF trial, a significant reduction in the incidence of ischemic stroke with warfarin (INR of 2.0 to 3.5) compared to aspirin (325 mg per day) has been shown only after four years of therapy. However, the rate of major bleeding events significantly increased throughout the treatment in the warfarin group (adjusted rate ratio 2.05 [95% CI 1.36-3.12], p<0.001).And:   
  30. The Cardiovascular Outcomes for People Using Anticoagulation Strategies (COMPASS) trial, investigating patients with systemic atherosclerotic disease, revealed a 49% reduction in the relative risk of stroke with rivaroxaban (2.5 mg twice daily) plus aspirin (100 mg once daily) compared to aspirin alone (100 mg once daily).
  31. Further:
  32. … Coronary Disease (COMMANDER HF) trial enrolling patients with HFrEF showed no benefit of rivaroxaban (2.5 mg twice daily) …
  33.  
  34.  
  35.  
  36. Lines 235-249: The reviewer would like to know if the authors are aware of any data on increase hemorrhagic risk in the trials mentioned and if they do then it should be included in this part of the discussion:Replay: We thank the reviewer for this important comment. We added these data to the discussion:Again, as in WATCH and WARCEF trials, the increased risk of major bleeding (1.68 [95% CI 1.18-2.39], p=0.003) was observed in the COMMANDER HF trial.And further regarding the COMPASS trial:However, the treatment with combined antiplatelet therapy resulted in 70% increase in major bleeding events in this patient cohort. 
  37.  
  38.  
  39.  
  40.  
  41.  
  42. Lines 261-64: The reviewer would recommend simplifying the language in the sentence since it’s quite confusing and difficult to comprehend  Recently, the Early Treatment of Atrial Fibrillation for Stroke Prevention Trial (EAST-AFNET 4) evaluated patients with AF from whom 28% had stable chronic HF. The findings of this study demonstrated an advantage of the early rhythm control therapy (treatment with antiarrhythmic drugs or AF ablation) compared to the symptomatic treatment of AF (usual care)
  43.  
  44. Replay: We thank the reviewer for this important comment. We rephrased the sentence now:
  45.  
